# Comparative Evaluation of Xanthan Gum, Guar Gum, and Scleroglucan Solutions for Mobility Control: Rheological Behavior, In-Situ Viscosity, and Injectivity in Porous Media

**DOI:** 10.3390/polym17131742

**Published:** 2025-06-23

**Authors:** Jose Maria Herrera Saravia, Rosangela Barros Zanoni Lopes Moreno

**Affiliations:** Post-Graduation Program in Petroleum and Science Engineering (CEP), School of Mechanical Engineering (FEM), Universidade Estadual de Campinas (UNICAMP), Campinas 13083-860, SP, Brazil; zanonilm@unicamp.br

**Keywords:** polymer flooding, biopolymers, mobility control, polymer injectivity

## Abstract

Water injection is the most widely used secondary recovery method, but its low viscosity limits sweep efficiency in heterogeneous carbonate reservoirs, especially when displacing heavy crude oils. Polymer flooding overcomes this by increasing the viscosity of the injected fluid and improving the mobility ratio. In this work, we compare three biopolymers (i.e., Xanthan Gum, Scleroglucan, and Guar Gum) using a core flood test on Indiana Limestone with 16–19% porosity and 180–220 mD permeability at 60 °C and 30,905 mg/L of salinity. We injected solutions at 100–1500 ppm and 0.5–6 cm^3^/min to measure the Resistance Factor (RF), Residual Resistance Factor (RRF), in situ viscosity, and relative injectivity. All polymers behaved as pseudoplastic fluids with no shear thickening. The RF rose from ~1.1 in the dilute regime to 5–16 in the semi-dilute regime, and the RRF spanned 1.2–5.8, indicating moderate, reversible permeability impairment. In-site viscosity reached up to eight times that of brine, while relative injectivity remained 0.5. Xanthan Gum delivered the highest viscosity boost and strongest shear thinning, Scleroglucan offered a balance of stable viscosity and a moderate RF, and Guar Gum gave predictable but lower viscosity enhancement. These results establish practical guidelines for selecting polymer types, concentration, and flow rate in reservoir-condition polymer flood designs.

## 1. Introduction

Primary recovery yields 10–20% of OOIP under depletion, and waterflooding can add another 20–40%; when waterflood efficiency declines, chemical EOR methods recover an additional 5–20% of oil [1,2,3,4,5]. Chemical EOR represents about 11% of all EOR projects worldwide, of which 77% employ polymer flooding and 23% use combined chemical–polymer schemes such as SP or ASP flooding [6,7,8]. Polymer flooding enhances sweep efficiency by increasing injected-fluid viscosity and improving the mobility ratio, making it especially effective in heterogeneous reservoirs where capillary forces and viscous fingering trap oil [7,9,10].

Polymer flooding (PF) is a well-established chemical EOR method that enhances sweep efficiency and delays water breakthrough by increasing the viscosity of the injection fluid relative to oil, thereby improving the mobility ratio in heterogeneous carbonate formations [11,12]. By adding high-molecular-weight, long-chain polymers to injection water, viscous fingering is mitigated and the cross-flow between high- and low-permeability zones is reduced [13]. The rheology of a polymer solution depends on factors such as polymer type, concentration, brine composition, and temperature; under reservoir flow conditions, these solutions typically exhibit thinning stress behavior indicative of pseudoplastic fluids [5,7,14].

Biopolymers such as Xanthan Gum, Scleroglucan, and Guar Gum offer several advantages over HPAM: they exhibit strong shear-thinning behavior that enhances injectivity [15], maintain viscosity stability at high shear rates and salinities up to ~30,000 mg/L TDS [5,6], resist thermal degradation above 80 °C [16,17], and adsorb less onto carbonate rock—38–78 lbm/acre-ft versus 35–1000 lbm/acre-ft for HPAM—thereby minimizing pore-throat blockage [5]. Additionally, the viscoelastic properties of these biopolymers help mitigate viscous fingering and optimize mobility control [13,18]. Comparative analyses show that, although Scleroglucan achieves optimal injectivity at lower polymer dosages, Xanthan Gum uniquely combines low adsorption with robust shear-thinning behavior, while Guar Gum offers a cost-effective option under moderate temperature and salinity conditions [6,17].

The evaluation of the polymer flooding potential requires comparing it to waterflooding in terms of injectivity, mobility control, and profitability [19]. The injection flow rate used in waterflooding can serve as a reference for polymer flooding [20]. Polymer flooding projects tend to be more profitable than waterflooding projects when they are evaluated across different scenarios to determine the best net present value [21]. However, field reports reveal a persistent reliance on synthetic polymers. Reports such as Renouf [22] documented only one biopolymer (Xanthan Gum) flood among thirty-two Canadian EOR projects, Standnes and Skjevrak [23] found just five biopolymer cases in sixty-two global polymer floods, many suffering injectivity losses of up to 80%. Moreover, the simulations of polymer injectivity often fail to match field outcomes, underscoring that near-wellbore polymer behavior remains poorly understood [20].

Injectivity loss can be classified into three categories: generic, anticipated, and undesired mechanisms [19]. The primary cause of injectivity reduction is the increase in viscosity due to polymer addition. Beyond viscosity effects, excessive polymer molecular weight or concentration can block pore throats and increase the pressure [19,24,25]; high inaccessible pore volume and fines migration damage permeability and reduce the effective flow area [7]; this results in low rock permeability [26], a poor mixture of polymers [27], and fines migration [28]. Suboptimal injection methods and well designs cause uneven fluid distribution [26]. Finally, poor injection water quality can destabilize the polymer solution and require extensive pretreatment [19]. Even minor salinity spikes can exacerbate this: for example, 2 wt % NaCl can reduce polymer solution viscosity by 23.8%, while 2 wt % Ca^2+^ can cause a ~28.6% drop [29]. Polymer retention further limits flood performance via hydrodynamic entrapment, irreversible adsorption onto clay and iron-rich surfaces, and mechanical trapping in pore throats narrower than their coil dimensions, delaying oil displacement and reducing sweep efficiency in carbonate reservoirs [30,31].

This study presents an in-depth analysis of fluid preparation, rheological behavior, and injectivity performance of Xanthan Gum, Scleroglucan, and Guar Gum solutions under reservoir-like salinity and temperature. The results characterize how polymer-specific flow curves and preparation protocols translate into injectivity responses across concentration and flow-rate regimes. The work is organized as follows: Section 2 describes the materials and experimental methodology; Section 3 presents the results and discussion from five core flooding experiments; Section 4 concludes with key findings.

## 2. Materials and Methods

Designing a successful core flooding experiment requires the careful consideration of several key factors and steps, as shown in Figure 1. These include selecting a representative core sample with known permeability and porosity property ranges, cleaning and preparing the sample before the experiment, choosing fluids compatible with the core that represent those used in field projects, controlling the flow rate and pressure of injected fluids, and designing the experimental setup. Lastly, the experiment should be structured to collect accurate and precise data, including the flow rates, pressures, and produced fluids [17].

### 2.1. Rock Samples

Indiana Limestone outcrops (20 cm length; 3.78 cm diameter) with a permeability range of 180–220 mD and a porosity range of 16–19% were used in a single-phase core flooding experiment. Routine core analysis (RCA) measured Gas Porosity and Gas Permeability as described in Section 2.3. Their average mineralogy are: calcium carbonate is 97.1%, magnesium carbonate is 1.2%, silica is 0.8%, alumina is 0.7%, iron oxide is 0.1%, and undetermined material is 0.1% [32]. Table 1 shows Indiana Limestone rock properties.

### 2.2. Fluid Preparations

Table 2 lists the composition of the synthetic seawater (SSW), which was formulated to simulate reservoir water during injection processes and has Total Dissolved Solids (TDS) of 30.905 mg/L [6,33,34]. Three commercial polymers were selected for the injectivity tests: Xanthan Gum and Guar Gum, supplied by Sigma Aldrich (Brazil), and Scleroglucan, provided by Biosynth Carbosynth (USA).

The direct dissolution of polymer powder into synthetic seawater can yield incomplete hydration, chain scission by divalent cations, and greater viscosity loss. Biopolymer stock solutions (4000 ppm) were prepared by dissolving the polymer powder in deionized water and stirring continuously at a speed sufficient to reach approximately 75% of the vortex height, ensuring full hydration and homogeneity [19,36,37]. To prevent microbial degradation, glutaraldehyde (50% *w*/*v*) was added as a biocide at a concentration of g/L immediately after hydration [17].

Each 4000-ppm stock solution was diluted with synthetic seawater, as shown in Table 2, to the target polymer concentration. The diluted solutions were then aged at 90 °C for the specified period, as shown in Table 3, prior to filtration.

The filtration process was performed using an 8 µm mesh, followed by a final filtration with a 1.2 µm mesh to retain bigger polymer chains. For Guar Gum, a pre-filtration step with ceramic filters was necessary due to excessive plugging in the mesh filters [17]; 55 µm, 20 µm, and 12 µm ceramic filters were used before passing through the 8 µm and 1.2 µm mesh filters, as shown in Table 3.

During filtration, the volume of filtrate and the time required to produce it were recorded, and the injection was 30 PSI. The filtration ratio $F_{r}$ was then calculated using Equation (1), as recommended in [38]. Previous work indicates that acceptable $F_{r}$ values for polymer solutions filtered through a 1.2 µm mesh range from 1.0 to 1.2 [19,39,40].(1)$$F_{r}=\frac{t_{500}-t_{400}}{t_{200}-t_{100}}$$
where *t**V*500, *t**V*400, *t**V*200, and *t**V*100 are the time required to filter volumes of 500, 400, 200, and 100 mL.

After filtration, the aged solution’s viscosity was measured both before and after the filtration process to quantify viscosity loss, which was calculated using Equation (2).(2)$$VL=\frac{\mu_{aging}-\mu_{1.2\mu mesh}}{\mu_{aging}}$$
where VL is the viscosity loss due to filtration, $\mu_{aging}$ is the viscosity after the period of aging, and $\mu_{1.2\mu mesh}$ is the viscosity reached after the filtration process.

The Power Law model shown in Equation (3) was employed to optimize the fitting of the flow curves observed in the polymer solutions.(3)$$\eta =K\;*\;{\dot{\gamma }}^{n-1}$$
where η is the apparent viscosity, $\dot{\gamma }$ is the shear rate, K is the flow consistency index, and n is the behavior index.

After filtration, each biopolymer solution was diluted with synthetic seawater to its target concentration. To remove entrained air, the solution was placed in a Kitassato flask (Sigma Aldrich, Brazil) connected to a vacuum pump and stirred for 10 min under vacuum. Once de-aerated, the polymer solution was carefully transferred into the injection accumulators.

### 2.3. Core Sample Preparation

Core flooding experiments begin by removing all original fluids from the core sample. The sample is placed in a Soxhlet extractor and cleaned with methanol. This solvent is heated to its boiling point (i.e., 64.7 °C) so that the vapor condenses in the core chamber, dissolving contaminants; the condensate is then siphoned back into the distillation flask. After two complete methanol extraction cycles, the core is dried at 80 °C for 24 h [41].

Once the rock sample was fully dry, it was placed into a core holder, and routine petrophysical measurements were performed using a gas porosimeter and Gas Permeability meter. The outlet of the core holder was then connected to a vacuum pump and held at 8 × 10^−3^ mbar for 24 h to evacuate all air. Next, a Mariotte bottle containing SSW brine was attached to the inlet, and the valve was opened so that brine imbibed the core under differential pressure, ensuring full saturation. The core was left to stabilize for one hour before the injection lines and pressure transducer were connected. Finally, brine was injected at the set flow rate while the outlet line was held closed against a backpressure valve set to 2000 PSI. This configuration maintained a constant downstream pressure of 2000 PSI, ensuring a stable pressure gradient across the sample throughout the test.

### 2.4. Methodology

In field applications, polymer injection often begins by incrementally increasing slug concentration until the desired viscosity is reached [20]. Following this strategy, our experiments inject three polymer concentrations at multiple flow rates to measure the Resistance Factor (RF), Residual Resistance Factor (RRF), in-site viscosity, and relative injectivity. Establishing the target polymer viscosity is thus the first step in designing a polymer flood, regardless of whether it is conducted in the lab, as a pilot test, or at full-field scale [42].

In the experimental setup shown in Figure 2, the sample was maintained at a constant temperature of 60 °C, while a back pressure valve was set at 2000 psi, and a confining pressure was set at 5000 psi. The injection sequence began with synthetic seawater (SSW) to determine the absolute permeability (kw), followed by the injection of polymer A (lowest concentration) up to polymer C (highest concentration). After each polymer injection, SSW was reintroduced.

A pore volume (PV) injection of 5 PV was initially set for each flow rate. However, this volume was reduced to 3 PV, as differential pressures stabilized rapidly during the first experiment.

Biopolymer solution concentrations were chosen to provide apparent viscosities 2.5 to 15 times that of brine (i.e., 0.48 cP at 60 °C and a shear rate of 10 s^−1^). The injection protocol comprises nine polymer stages alternated with five brine stages, each run at a specific flow rate. To avoid flow-rate surges when switching between SSW and polymer, an extra pore volume of brine was injected at the end of each brine stage to reset the flow rate for the next polymer stage.

In our core flood experiments, the pressure was monitored via Rosemount pressure transducer (Emerson, USA) (0–9 PSI and 0–36 PSI ranges) and Cristall pressure sensors (Ametek, USA) (0–15,000 PSI), all connected to taps positioned along the core holder. The exact spacing and location of each pressure tap are shown in Figure 2. These detailed pressure profiles reveal fluid velocities, pressure drops, and flow patterns within the porous medium, which are critical for evaluating displacement efficiency and calibrating numerical flow models.

The Resistance Factor (RF) is a key parameter in assessing the efficacy of polymer injection in core flooding experiments. It is defined by Equation (4) as the ratio of the differential pressure required to inject a polymer solution to that required for water injection. The RF is influenced by several parameters, including viscosity, adsorption, and retention [5,43,44].(4)$$RF=\frac{\lambda_{w}}{\lambda_{p}}=\frac{{\left(\frac{k}{\mu }\right)}_{wf}}{{\left(\frac{k}{\mu }\right)}_{pf}}=\frac{{\Delta P}_{pf}}{{\Delta P}_{wf}}$$
where λ, k, and μ represent the phase mobility, effective permeability, and viscosity, respectively. RF refers to the Resistance Factor, and ΔP refers to the differential pressure. The subscripts “wf” and “pf” denote the water and polymer flooding, respectively.

The permeability reduction in core flooding experiments can be assessed using the Residual Resistance Factor (RRF). As seen in Equation (5), this parameter quantifies the change in water mobility resulting from the rock surface’s exposure to the polymer, representing the variation in core permeability before and after polymer injection [5,43].(5)$$RRF=\frac{{\left(\frac{k}{\mu }\right)}_{before-pf}}{{\left(\frac{k}{\mu }\right)}_{after-pf}}=\frac{{\Delta P}_{w\;after-pf}}{{\Delta P}_{w\;before-pf}}$$
where k and μ represent the effective permeability and viscosity, respectively. RRF refers to the Residual Resistance Factor, and ΔP refers to the differential pressure. The subscript “pf” denotes polymer flooding.

A high RRF indicates a reduction in water injectivity. Conversely, an RRF value of one suggests no change in rock permeability, which would be counterproductive as it indicates no permeability reduction. The RRF depends on the type and concentration of the polymer, as well as the amount of polymer retained within the rock [8,39,45].

Figure 3 shows the typical pressure drop in the core for a polymer core flooding experiment. One approach to determining the apparent viscosity of the polymer within the porous medium is by calculating the ratio between the RF and the RRF, as expressed in Equation (6). The apparent viscosity characterizes the macroscopic rheological behavior of the polymer solution in the porous medium [10,46].(6)$$\mu_{app}=\frac{RF}{RRF}*\mu_{w}$$
where μ represent the viscosity. RF and RRF refer to the Resistance Factor and Residual Resistance Factor, respectively. The subscript “w” refers to the water phase.

Polymer injectivity loss is quantified by the relative injectivity ($I_{r}$), defined as the ratio of polymer to water infectivity under identical flow conditions. To calculate $I_{r}$, the injection flow rate or interstitial velocity and the corresponding differential pressures for both brine and polymer injections are recorded and applied in Equation (7). Acceptable $I_{r}$ values typically lie between 0.5 and 0.9, indicating moderate injectivity impairment [20,25,48].(7)$$I_{r}=\frac{I_{p}}{I_{b}}=\frac{{\Delta P}_{wf}}{{\Delta P}_{pf}}$$
where $I_{r}$ represents relative injectivity, and $I_{p}$ and $I_{b}$ denote polymer injectivity and brine injectivity. ΔP refers to the differential pressure. The subscripts “wf” and “pf” denote the water and polymer flooding, respectively.

## 3. Results and Discussion

The filtration test is a crucial step in core flooding experiments, as it ensures solution integrity and prevents pore-throat blockage [17,47]. In this study, we used a Haake Mars III rheometer to measure the rheological properties of each polymer solution before and after filtration, quantifying the resulting viscosity loss.

Figure 4a presents the viscosity results for Xanthan Gum solutions before and after filtration. At a reference shear rate of 10 s^−1^, viscosity losses were 31.2% for IL-45 and 37.3% and 37.5% for IL-46 and IL-34, respectively. These losses occur because the filter mesh selectively retains the highest-molecular-weight polymer coils (i.e., which are more numerous and entangled at higher concentrations), leaving behind a solution of short chains with lower average molecular weight and thus lower viscosity. The nearly identical losses for IL-46 and IL-34 arise from their similar concentrations, with minor differences attributable to slight variations in handling or light exposure.

Figure 4b compares Guar Gum (GG) and Scleroglucan (SCLG) viscosity solutions prior to and post filtration. The GG solution experienced the highest viscosity loss, around 75.1%, at a reference shear rate of 10 s^−1^. This suggests that a large fraction of its thickening power derives from insoluble fibers and proteins removed by the filter. In contrast, the SCLG solution exhibited the lowest loss with 26.6% at the reference shear rate, reflecting its high solubility and structural stability during filtration. Nevertheless, all biopolymer solutions demonstrated a good filtration performance, with filtration ratios $F_{r}$ of about 1.03 when using a 1.2 µm mesh, as calculated by Equation (1).

Figure 5a,c show the Resistance Factor (RF) of Xanthan Gum (XG) solutions as a function of the polymer concentration. In the dilute regime, below the Critical Concentration (C*), the Resistance Factor remains close to unity, indicating near-Newtonian flow with minimal viscosity effects. Once concentrations enter the semi-dilute regime (700, 1100, and 1500 ppm), the RF rises sharply: IL-45 reaches 5.6 at 700 pp, IL-34 hits 6.4 at 1100 ppm, and both climb to 15.9 at 1500 ppm. The high Resistance Factor values reflect increased polymer–rock interactions (e.g., adsorption and pore-throat restriction), while the decline in the RF at higher flow rates demonstrates shear thinning as polymer chains align under shear stress.

Figure 5b,d shows the Residual Resistance Factor (RRF) of Xanthan Gum (XG) solutions, which ranges from 1.8 down to 1.2 in the dilute regime and from 2.6 to 2.0 in the semi-dilute regime for XG, indicating only moderate, partially reversible permeability reduction.

For the IL-33-SCLG case, Figure 5e shows a drastic increase in the Resistance Factor at 800 ppm, reaching values of up to 14.1 at lower injection rates. This suggests that at higher concentrations, the Scleroglucan solution forms entangled polymer networks, significantly increasing flow resistance. The sharp rise in the Resistance Factor highlights the strong viscoelastic properties of Scleroglucan in this concentration range, which is characteristic of biopolymer solutions exhibiting shear-thinning behavior.

Figure 5e,f shows that at 800 ppm, the Resistance Factor of the IL-33 Scleroglucan solution soars to 14.1 under the lowest flow rate (i.e., evidence of highly entangled, viscoelastic networks and pronounced shear thinning), while the Residual Resistance Factor peaks at just 5.8 before falling to 3.3 at higher rates. This disparity indicates that, although Scleroglucan greatly increases flow resistance during injection, most of the permeability impairment is reversible upon brine flushing, likely due to polymer chain detachment or redistribution within the pore space.

Figure 5g,h compare the Resistance Factor (RF) and Residual Resistance Factor (RRF) for the IL-37 Guar Gum solution. In the dilute regime (i.e., 100–300 ppm), the RF remains low and stable, ranging from 1.3 to 1.3, indicating minimal increases in viscosity or flow resistance. Upon entering the semi-dilute regime (i.e., 1200 ppm), the RF rises to approximately 2.3 but is still lower than values observed with Xanthan Gum and Scleroglucan solutions. This suggests that Guar Gum forms a less rigid network in solution. The corresponding RRF values follow a similar trend, ranging from 1.1 to 1.3 in the dilute regime and 2.1 to 1.3 in the semi-dilute regime, showing that much of the polymer-induced permeability reduction is reversible upon brine flushing. The further decline in the RRF at higher flow rates implies a weak retention of Guar Gum chains in the porous medium, reinforcing that higher concentrations are needed to achieve comparable viscosity effects to other biopolymers.

Figure 6 presents the Resistance Factor (RF) and Residual Resistance Factor (RRF) for the IL-34 core flooded with Xanthan Gum at concentrations of 100, 700, and 1100 ppm, with flow rates of up to 14 cm^3^/min. The IL-34 core was selected for its highest permeability (i.e., an optimal strategy for biopolymer flooding at higher flow rates). At 700 ppm, the RF rises to 6.5 at 1 cm^3^/min and declines to 2.8 at 14 cm^3^/min, exceeding IL-45’s RF at the same concentration. At 1100 ppm, the RF increases to 9.7 at 1 cm^3^/min and decreases to 4.0 at 14 cm^3^/min. Overall, the RF drops by about 60% as the flow rate increases from 1 to 14 cm^3^/min, with smaller reductions at higher rates due to reduced polymer–rock interactions. Conversely, the RRF peaks at 3.3 for 700 ppm and 3.0 for 1100 ppm at 1 cm^3^/min, then declines to 1.8–2.1 at 14 cm^3^/min. This behavior indicates that mechanical trapping dominates retention under high shear rates and that part of the permeability impairment is reversible upon brine flushing.

Figure 7 compares the in situ viscosity ($\mu_{app}$) of all five core–polymer combinations as a function of flow rate and concentration. In the IL-45 Figure 7a and IL-46 Figure 7d cores, Xanthan Gum solutions show pronounced shear thinning: ($\mu_{app}$) drops from 3 m·Pas at 1 cm^3^/min down to brine baseline (i.e., 0.5 mPa·s) by 14 cm^3^/min. In contrast, the high-permeability IL-34 core Figure 7e exhibits only moderate thinning: its 100 ppm solution remains near the brine, and even at 1100 ppm, the ($\mu_{app}$) peaks at 1.6 mPa·s before declining more gently; this suggests that larger pore channels generate lower shear rates and promote more uniform displacement.

The Guar Gum solution in Figure 7b shows an almost flat apparent viscosity curve across all rates, reflecting its flexible, low-molecular-weight chains that reconfigure easily. This behavior yields excellent injectivity but minimal mobility control. Scleroglucan in Figure 7b sits between these extremes (i.e., its branched network provides moderate shear thinning, a good sweep improvement, with manageable injection pressure.

These behaviors underscore that optimal polymer selection depends on matching the rheological profile to reservoir permeability; strong shear-thinning solutions like Xanthan Gum excel at mobility control in tight rocks but may lose viscosity and risk channeling at high rates; flexible gums like Guar Gum solutions maintain injectivity but afford limited sweep improvement; intermediate polymers, such as the Scleroglucan solution, offer a compromise for broad field conditions.

Table 4 presents the behavior index (n) and consistency coefficient (K) derived from the apparent viscosity data shown in Table 3. Xanthan Gum exhibits an increase in viscosity and more pronounced shear-thinning behavior with rising polymer concentrations. This is evidenced by the increase in the consistency coefficient and the concurrent decrease in the flow behavior index. For instance, in IL-45, K increases from 0.489 at 50 ppm to 1.142 at 700 ppm, while n decreases from 0.995 to 0.868, indicating a transition from near-Newtonian to pseudoplastic behavior. Similarly, in IL-34, K rises from 0.666 at 100 ppm to 2.010 at 1100 ppm, accompanied by a drop in n from 0.952 to 0.718, reflecting strong shear-thinning characteristics. These properties make Xanthan Gum particularly suitable for applications requiring high viscosity under low-shear conditions, such as mobility control in enhanced oil recovery (EOR) operations.

Guar gum (GG) exhibits relatively stable consistency coefficients along with a mild shear-thinning tendency. In the IL-37 case, the consistency coefficient (K) remains within a narrow range—from 0.473 to 0.760—while the flow behavior index (n) stays close to 1.0, indicating near-Newtonian behavior. This rheological profile suggests that GG maintains consistent flow properties across varying shear rates, which may be advantageous in applications that demand stable viscosity and less shear-dependent behavior.

Scleroglucan exhibits increasing viscosity and more pronounced shear-thinning behavior with a rising concentration, as reflected by the increase in the consistency coefficient (K) from 0.560 at 100 ppm to 1.007 at 800 ppm. Simultaneously, the flow behavior index (n) decreases from 0.996 to 0.859, as shown in Table 4. These trends indicate a transition from nearly Newtonian to more distinctly pseudoplastic behavior, highlighting the enhanced rheological response of Scleroglucan at higher concentrations.

The relative injectivity, derived from Equation (7), compares the pressure drops of polymer solutions to those observed during the initial brine injection. As shown in Figure 8a, increasing the injection flow rate enhances the relative injectivity in the semi-dilute regime. For IL-45 using Xanthan Gum at 50, 100, and 700 ppm, the small differences in injectivity between the 100 ppm and 700 ppm concentrations may be attributed to the initial plugging of the core surface, which could result in irregular injection behavior.

The IL-46 Figure 8d results are consistent with the rheological behavior of Xanthan Gum, where higher concentrations lead to increased viscosity and flow resistance, ultimately reducing injectivity. Additionally, stronger fluid–rock interactions at elevated concentrations contribute to the observed decline in relative injectivity. For IL-34, the most permeable core in this study, Xanthan Gum solutions at 700 ppm and 1100 ppm exhibited similar injectivity values, indicating a potential optimal concentration range beyond which further increases provide no additional benefit. The slight increase observed at higher concentrations may suggest polymer chain degradation or shear-induced breakage. These findings highlight the critical role of core permeability in determining injectivity performance.

On the other hand, the Guar Gum (GG) solution Figure 8b behaves almost as a Newtonian fluid, maintaining stable injectivity across increasing flow rates and exhibiting a low resistance to flow. However, at higher concentrations, such as 1200 ppm, relative injectivity decreases notably, likely due to increased viscosity and potential polymer retention. Despite this, GG remains suitable for applications requiring consistent injectivity over a wide range of flow rates, especially in formations with moderate permeability. Compared to other biopolymers, GG demands much higher concentrations to achieve comparable viscosities, which may raise concerns regarding cost and material consumption. Nevertheless, its predictable and stable behavior offers precise control over viscosity levels, making it advantageous in scenarios where flow behavior must remain consistent.

Figure 8c shows that as the concentration of Scleroglucan increases, the viscosity of the solution rises accordingly, leading to higher pressure drops and lower relative injectivity. This is expected, as greater polymer concentrations typically result in increased flow resistance. However, in situ viscosity measurements show similar viscosities for the 500 ppm and 800 ppm solutions, which might appear contradictory at first glance. It is important to consider that in situ viscosity is affected by factors such as the shear rate and pore space configuration, which may vary throughout different sections of the rock core. These local variations can lead to discrepancies between expected rheological behavior and actual performance during core flooding experiments.

Beyond the lab-scale trends, these Resistance Factor (RF) and Residual Resistance Factor (RRF) flow-rate relationships carry direct implications for field-scale injectivity design. In the near-wellbore region, high shear rates prevail, so choosing a polymer and concentration that delivers sufficient shear thinning will minimize pressure buildup without compromising sweep. For example, IL-45 and IL-46 with Xanthan Gum solution tests show that at field-equivalent superficial velocities (1–5 m/d), the RF can be held below critical limits (RF < 5) by operating in the semi-dilute regime and exploiting shear alignment. Conversely, in low-permeability zones where shear rates drop, a higher RF bolsters mobility control, improving sweep. Thus, by mapping lab flow rates to reservoir velocities, one can select the polymer dose and injection rate that balance mobility contrast against injectivity, avoiding near-wellbore plugging while still enhancing sweep efficiency.

## 4. Conclusions

Under reservoir-like conditions (i.e., 60 °C and 30,905 mg/L TDS), core flood experiments demonstrate that each biopolymer’s unique rheological profile and preparation protocol directly dictate its injectivity performance. Xanthan Gum Solutions shows the strongest pseudoplastic behavior, yielding a high Resistance Factor of up to 16 but a sharp apparent viscosity drop at high shear rates; Scleroglucan combines a moderate Resistance Factor of up to 8 with more stable apparent viscosity; Guar Gum delivers a nearly constant apparent viscosity with a minimal injectivity loss.

These results characterize how polymer-specific flow curves and preparation methods translate into injectivity responses across concentration and flow-rate regimes. By matching rheological behavior to reservoir permeability and desired injection conditions, an optimal polymer type, concentration, and injection rate to maximize sweep efficiency without compromising injectivity can be selected.

## Figures and Tables

**Figure 1 polymers-17-01742-f001:**
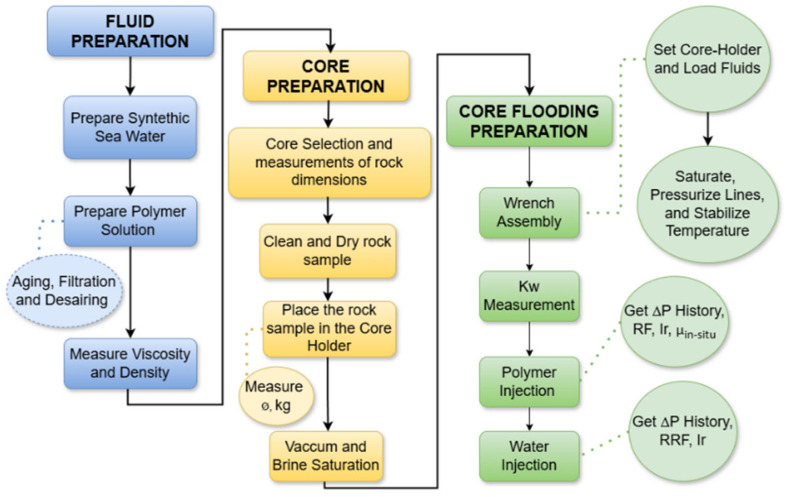
Experiment workflow.

**Figure 2 polymers-17-01742-f002:**
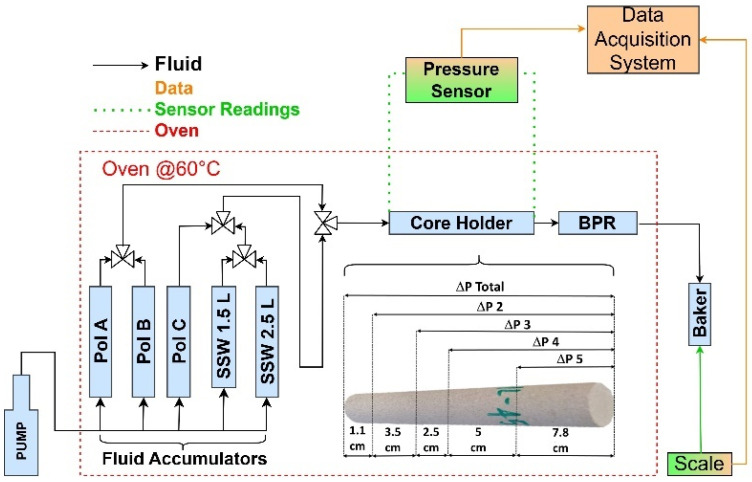
Experimental setup [35].

**Figure 3 polymers-17-01742-f003:**
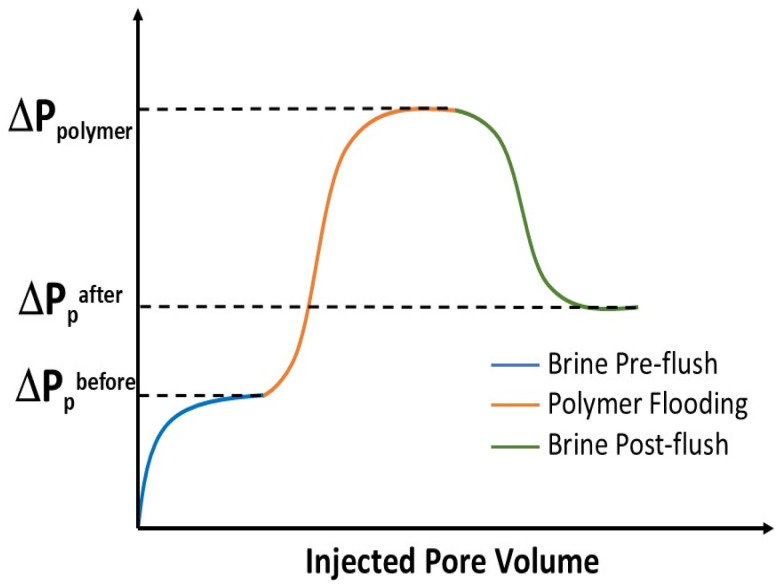
Typical pressure drops in the core for polymer flooding [47].

**Figure 4 polymers-17-01742-f004:**
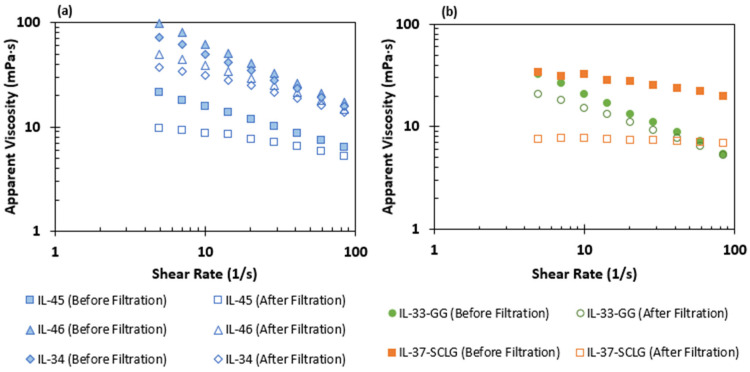
Viscosity loss due to the filtration process (**a**) from Xanthan Gum solutions and (**b**) Scleroglucan and Guar Gum solutions.

**Figure 5 polymers-17-01742-f005:**
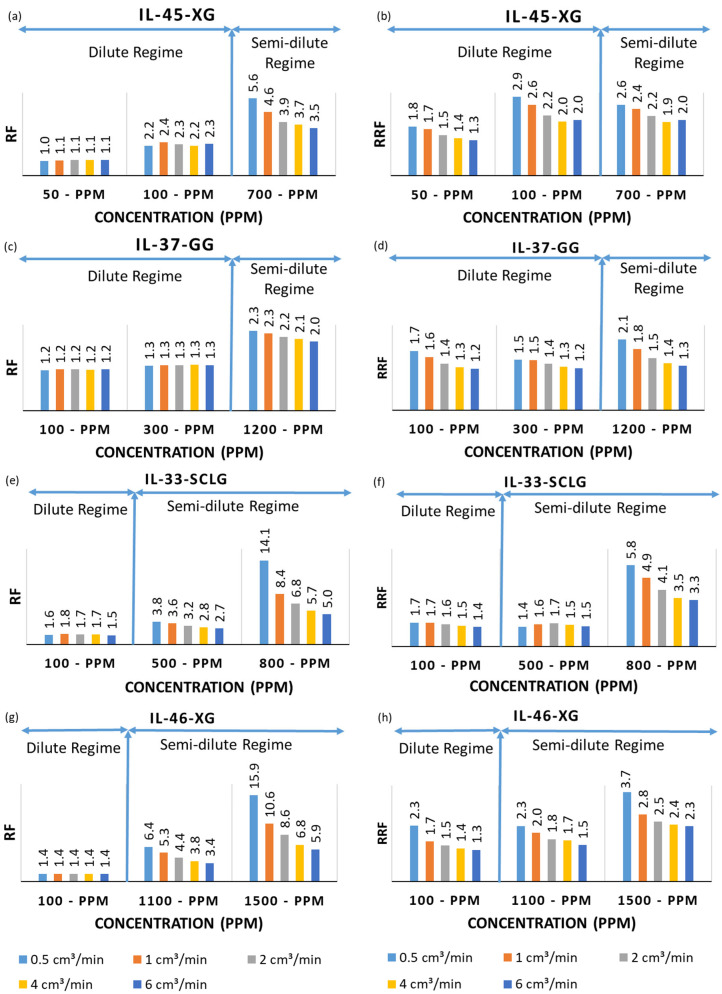
Resistance Factor and Residual Resistance Factor results from the biopolymer solutions, (**a**) RF result from IL-45 of Xanthan Gum test; (**b**) RRF result from IL-45 Xanthan Gum test; (**c**) RF result from IL-37 Guar Gum test; (**d**) RRF result from IL-37 Guar Gum test; (**e**) RF result from IL-33 Scleroglucan test; (**f**) RRF result from IL-33 Scleroglucan test; (**g**) RF result from IL-45 Xanthan Gum test; (**h**) RRF result from IL-45 Xanthan Gum test.

**Figure 6 polymers-17-01742-f006:**
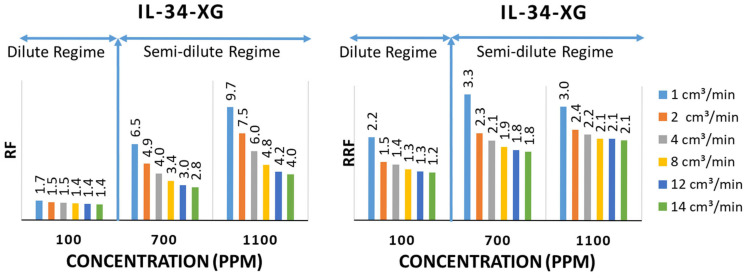
Resistance Factor and Residual Resistance Factor results from IL-34 [35].

**Figure 7 polymers-17-01742-f007:**
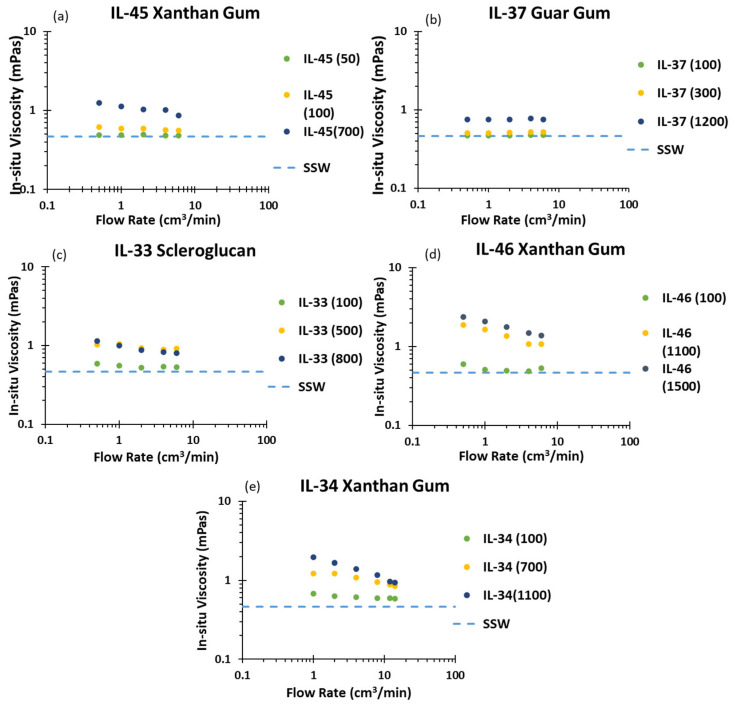
In situ viscosity results (**a**) In situ viscosity from IL-45 Xanthan Gum test; (**b**) In situ viscosity from IL-37 Guar Gum; (**c**) In situ viscosity from IL-33 Scleroglucan test (**d**) In situ viscosity from IL-46 Xanthan Gum test (**e**) In situ viscosity from IL-34 Xanthan Gum test.

**Figure 8 polymers-17-01742-f008:**
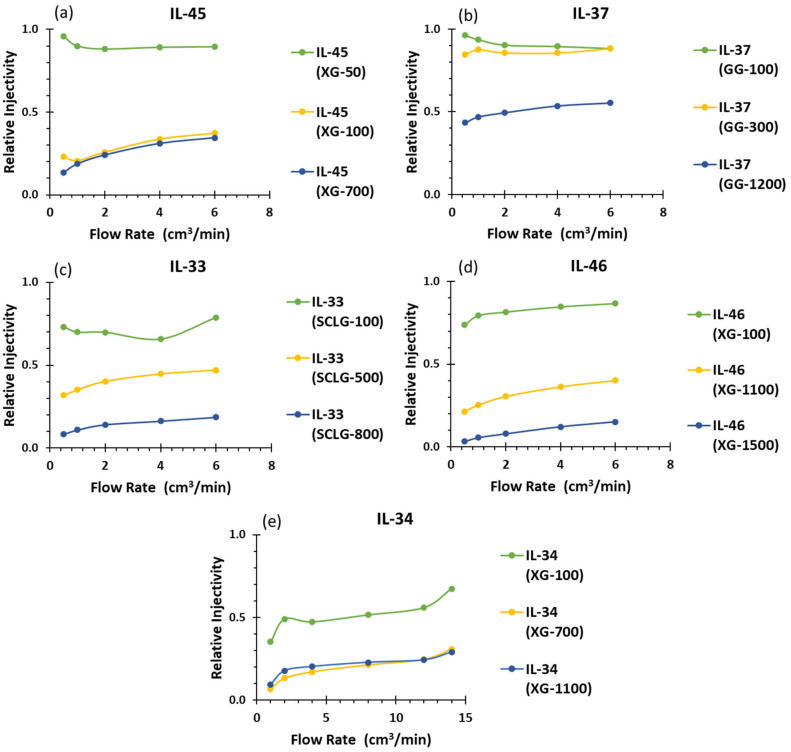
Relative Injectivity results (**a**) $I_{r}$
from IL-45 Xanthan Gum test (**b**) $I_{r}$ from IL-37 Guar Gum test (**c**) $I_{r}$ from IL-33 Scleroglucan test (**d**) $I_{r}$ from IL-46 Xanthan Gum test (**e**) $I_{r}$ IL-34 Xanthan Gum test.

**Table 1 polymers-17-01742-t001:** Indiana Limestone core sample properties.

Rock/Polymer Code	Gas Porosity (%)	Gas Permeability (mD)	Length (cm)	Diameter (cm)	Mass (g)
IL-45-XG	17.59	173.3	19.9	3.78	499.9
IL-37-GG	17.95	171.2	19.9	3.78	500.2
IL-33-SCLG	17.59	142.6	19.9	3.78	502.9
IL-46-XG	16.50	160.1	19.9	3.78	501.9
IL-34-XG	18.38	639.7	20.0	3.78	501.5

**Table 2 polymers-17-01742-t002:** Synthetic sea water (SSW) composition [35].

Composition	Chemical Formula	Concentration 100% (ppm)
Potassium Chloride	KCl	749.3
Calcium Chloride Dihydrate	CaCl_2_ 2H_2_O	484.2
Magnesium Chloride Hydrate	MgCl_2_ H_2_O	1271.3
Strontium Chloride Hexahydrate	SrCl_2_ 6H_2_O	5.2
Barium Chloride Hexahydrate	BaCl_2_ 2H_2_O	2.0
Lithium Chloride	LiCl	1.2
Sodium Bromide	NaBr	82.4
Sodium Sulfate	Na_2_SO_4_	57.7
Sodium Chloride	NaCl	28,252.2
Total TDS		30,905.5

**Table 3 polymers-17-01742-t003:** Summary of core flooding preparation experiments.

Parameter\Code	IL-45	IL-46	IL-34	IL-33	IL-37
Polymer	Xanthan Gum	Xanthan Gum	Xanthan Gum	Scleroglucan	Guar Gum
Overlap Concentration C * (ppm)	285	285	285	175	950
Biocide	Yes	Yes	Yes	Yes	Yes
Agitation	1 day	1 day	1 day	7 days	1 day
Dissolution Before Aging (ppm)	1000	2000	2000	1000	3000
Aging Time	1 day	1 day	1 day	1 day	3 h
Filtration Mesh	8 µm 1.2 µm	8 µm 1.2 µm	8 µm 1.2 µm	8 µm 1.2 µm	55 µm * + 20 µm * + 12 µm * + 8 µm + 1.2 µm
Filtration Ratio at 1.2 Mesh	1.02	1.03	1.04	1.05	1.02
Concentration Test (ppm)	50	100	100	100	100
	100	1100	700	500	300
	700	1500	1100	800	1200
Viscosity (cP) at 10 s^−1^	0.56	0.66	0.66	0.73	0.49
	0.67	10.71	3.73	4.46	0.62
	4.40	20.51	8.51	9.81	2.30
Kw (mD)	127.9	156.7	476.4	147.1	156.4

^*^ ceramic filter.

**Table 4 polymers-17-01742-t004:** Power Law coefficients.

Test	Concentration	K	n	R^2^
IL-45	XG-50	0.489	0.995	0.216
IL-45	XG-100	0.597	0.962	0.932
IL-45	XG-700	1.142	0.868	0.934
IL-37	GG-100	0.509	1.015	0.951
IL-37	GG-300	0.473	1.004	0.459
IL-37	GG-1200	0.760	1.003	0.149
IL-33	SCLG-100	0.560	0.996	0.715
IL-33	SClLG-500	1.001	0.933	0.806
IL-33	SCLG-800	1.007	0.859	0.959
IL-46	XG-100	0.542	0.948	0.435
IL-46	XG-1100	1.599	0.757	0.983
IL-46	XG-1500	2.057	0.776	0.995
IL-34	XG-100	0.666	0.952	0.936
IL-34	XG-700	1.293	0.851	0.928
IL-34	XG-1100	2.010	0.718	0.990

## Data Availability

Data are contained within the article.

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
