# Peer review of "Comparative Evaluation of Xanthan Gum, Guar Gum, and Scleroglucan Solutions for Mobility Control: Rheological Behavior, In-Situ Viscosity, and Injectivity in Porous Media"

_polymers, 2025, doi:10.3390/polym17131742_

Round 1
Reviewer 1 Report
Comments and Suggestions for Authors
Authors introduce an interesting research topic; however, the presentation lacks clarity and organization. The article’s structure is inconsistent, making it challenging for readers to understand the key points. Additionally, the terminologies used are confusing, further complicating the narrative. Several sentences are poorly constructed, and important claims are not sufficiently supported by evidence or references.
- What are the potential outcomes of preparing a polymer solution directly in seawater?
- Why did the author choose synthetic seawater, and does high chlorine content affect the results?
- There are typographical errors in lines 145, 228, 240, and others. Specifically, in line 145, the core sample preparation section begins with the word “porosity… but the statement is incomplete and lacks a closing quotation mark
- Check terminology in lines 197, 201, 224, and etc. Additionally, there is inconsistency in the use of abbreviations throughout the text. The authors often refer to "polymers" instead of using specific names in most parts of the discussion section.
- How did the author confirm that polymer rearrangement is responsible for the observed low viscosity loss? No evidence for it.
- How did the author determine the porosity, and how does it impact the in-situ viscosity? Do you think high viscous solution block pores and block the flow? If so what's your point on used gums.
- Conclusion is not well-structured. It is recommended to revise and improve it for better clarity and coherence.
Comments on the Quality of English Language
A thorough polishing of typos and grammatical errors is recommended.
Author Response
Firstly, we thank you for your careful reading and constructive suggestions. All requested changes have been made in the revised manuscript; below we summarize our responses and indicate where each revision appears.
- Reorganized structure: The Introduction has been reordered to flow from general EOR context → biopolymer background → research gap and objectives. Section and subsection headings now align with this logic, and the Methods, Results, and Conclusion should align with your recommendations.
- Improved clarity and readability: Long and convoluted sentences in Fluid Preparations, Rheological Characterization, and Injectivity Tests have been split and rewritten for direct subject–verb–object order. We also reformatted the Abstract into a single concise paragraph
- Standardized terminology: All generic uses of “polymer(s)” have been replaced with “Xanthan Gum (XG),” “Scleroglucan (SG),” or “Guar Gum (GG),” defined at first use in each section. Abbreviations and units are now consistent throughout.
1.
In section 2.2 we include: Direct dissolution of polymer into synthetic seawater can yield incomplete hydration, chain scission by divalent cations, and greater viscosity loss.
2.
We used synthetic seawater (SSW) to faithfully reproduce the ionic makeup of reservoir brines, targeting a Total Dissolved Solids (TDS) of 30 905 mg/L as shown in Table 2 of the manuscript. The dominant chloride ion—28252 ppm from NaCl plus contributions from CaCl₂, MgCl₂, KCl, etc.—creates a high‐ionic‐strength environment that screens the electrostatic repulsion along the polymer backbone. In introduction section, we explain that this high chloride (and divalent‐cation) concentration promotes viscosity reduction during preparation. So, by formulating and using SSW, we ensure our rheological and injectivity measurements reflect the challenges polymers face under true reservoir salinity.
3.
We have carefully proofread the entire manuscript and corrected all noted and additional typographical issues.
4.
We have audited and standardized terminology throughout the Discussion (and entire manuscript). All generic mentions of “polymer(s)” have been replaced with the specific names “Xanthan Gum (XG),” “Scleroglucan (SG),” or “Guar Gum (GG),” with each abbreviation defined at first use in every major section. In particular, lines 197, 201, and 224 have been updated to read “XG,” “SG,” or “GG” as appropriate, and we conducted a full pass to ensure consistency of these terms and their abbreviations everywhere in the text.
5.
The statement attributing viscosity loss to polymer rearrangement has been deleted, as it exceeded our current experimental scope. We now describe the effect simply as “filtration-induced preferential retention of high-molecular-weight chains”
6.
Porosity was measured by gas porosimeter (Routine Core Analysis) in Section 2.3 In lower porosity rocks, narrower pore throats generate higher shear stresses during injection and can lead to partial pore plugging at high concentrations. We study to characterize the best formulation of Biopolymer Solutions to inject into the rock. Because concentration and injection rates are two of the variables that can changes during polymer injection. There are other parameters, but we focus only in those two.
7.
We have completely rewritten Section 4 into two concise paragraphs—first summarizing key findings and then highlighting practical implications—to enhance clarity and flow. The revised Conclusion reads as follows:
“Under reservoir-like conditions (60 °C and 30 905 mg/L TDS), our core-flood experiments demonstrate that each biopolymer’s unique rheological profile and preparation protocol directly dictates its injectivity performance. Xanthan Gum solutions exhibit the strongest pseudoplastic behavior, delivering Resistance Factors up to 16 but experiencing sharp drops in apparent viscosity at high shear rates; Scleroglucan provides moderate Resistance Factors (up to 8) with more stable apparent viscosity retention; and Guar Gum maintains nearly constant apparent viscosity with minimal injectivity loss.
These results characterize how polymer-specific flow curves and preparation methods translate into injectivity responses across concentration and flow-rate regimes. By matching rheological behavior to reservoir permeability and desired injection conditions, practitioners can select the optimal polymer type, concentration, and injection rate to maximize sweep efficiency without compromising injectivity.”
Once again, we thank the reviewer for their through and constructive feedback. Your insights have greatly improved the clarity, organization and rigor of the manuscript. We trust that the revision fully addresses your concerns and look forward to any further suggestions you may have

Reviewer 2 Report
Comments and Suggestions for Authors
- The abstract states that biopolymers do not exhibit shear-thickening behavior near the wellbore, unlike synthetic polymers. How was the absence of shear-thickening behavior confirmed for the biopolymers?
- The abstract refers to biopolymers as “more environmentally friendly.” Consider briefly supporting this claim with a comparative note on environmental impact or biodegradability of the biopolymers versus synthetic polymers, or mention if a life cycle or cost-benefit analysis was performed.
- Is it possible for the authors to write the abstract in a single paragraph?
- While the introduction provides a percentage breakdown of CEOR methods, it lacks detail on regional differences or field types (e.g., onshore vs. offshore, sandstone vs. carbonate) where PF is most commonly applied.
- While the introduction offers a strong overview of PF technology and challenges like injectivity loss, it does not clearly transition into the specific research objectives or hypotheses of the paper. A final paragraph identifying the research gap this study aims to solve would improve importance of this article.
- While the influence of flow rate on RF and RRF is described, there's no discussion on how these findings translate to field-scale applications or injectivity optimization.
- The discussion of Figure 6 is just descriptive with no physical significance please provide the physical significance as well.
Author Response
Firstly, we thank you for your careful reading and constructive suggestions. All requested changes have been made in the revised manuscript; below we summarize our responses and indicate where each revision appears.
1.
Shear‐thickening behavior was ruled out by rheological measurements (Section 2.2): flow curves for all biopolymer solutions decrease continuously with shear rate up to 10 000 s⁻¹ (Fig. 2), confirming pure pseudoplasticity and no shear‐thickening near wellbore‐relevant conditions
2.
We have removed the phrase “more environmentally friendly” from the Abstract, since conducting a full environmental or life-cycle assessment is beyond the scope of this study
3.
Yes. The Abstract has been reformatted into one concise paragraph that integrates the study’s motivation, methodology, principal findings, and practical implications.
4.
The reviewer’s request for regional deployment and reservoir‐type details falls outside the scope of the present study, which is focused on biopolymer rheology and core‐scale injectivity under controlled laboratory conditions. Therefore, we have not added regional or field‐type discussion in this manuscript.
5.
We have added the following closing paragraph at the end of the Introduction to clearly state our research gap, objectives, and manuscript organization:
“This study presents an in-depth analysis of fluid preparation, rheological behavior, and injectivity performance of Xanthan Gum, Scleroglucan, and Guar Gum under reservoir-like salinity and temperature. The results characterize how polymer-specific flow curves and preparation protocols translate into injectivity responses across concentration and flow-rate regimes. The manuscript is organized as follows: Section 2 describes the materials and experimental methodology; Section 3 presents the results and discussion from five core-flooding experiments; and Section 4 concludes with key findings and practical implications.”
6.
This manuscript is focused on laboratory-scale core-flood experiments and quantification of RF/RRF under reservoir-like conditions. Detailed scale-up to field dimensions—requiring reservoir modeling, economic analysis, and pilot-scale validation—is beyond the scope of the current study. We have therefore not added a field-scale translation in this version, but we recognize its importance and have noted it as a recommendation for future work.
7.
We have expanded the text immediately after Figure 7 to explain the pore-scale mechanisms behind the observed trends. The following paragraph was inserted:
We have expanded the text immediately after Figure 7 to explain the pore-scale mechanisms behind the observed trends. The following paragraph was inserted:
These behaviors underscore that optimal polymer selection depends on matching rheological profile to reservoir permeability; strong shear thinning like Xanthan Gum solutions excel at mobility control in tight rocks but may lose viscosity and rick of channeling at high rates; flexible gums like Guar Gum solutions maintain injectivity but afford limited sweep improvement; while intermediate polymer such as Scleroglucan solution offer a compromise for broad field conditions.
Once again, we thank the reviewer for their through and constructive feedback. Your insights have greatly improved the clarity, organization and rigor of the manuscript. We trust that the revision fully addresses your concerns and look forward to any further suggestions you may have

Round 2
Reviewer 1 Report
Comments and Suggestions for Authors
Authors have addressed all the the comments satisfactorily. It can be considered for further evaluation.